# Use of the Hospital Survey of Patient Safety Culture in Norwegian Hospitals: A Systematic Review

**DOI:** 10.3390/ijerph18126518

**Published:** 2021-06-17

**Authors:** Espen Olsen, Ann-Chatrin Linqvist Leonardsen

**Affiliations:** 1Department of Innovation, Leadership and Marketing, UiS Business School, University of Stavanger, 4036 Stavanger, Norway; 2Department of Health and Welfare, Ostfold University College, 1757 Halden, Norway; ann.c.leonardsen@hiof.no

**Keywords:** patient safety, patient safety culture, measurement quality, health services, review

## Abstract

This review aims to provide an overview of empirical studies using the HSOPSC in Norway and to develop recommendations for further research on patient safety culture. Oria, an online catalogue of scientific databases, was searched for patient safety culture in February 2021. In addition, three articles were identified via Google Scholar searches. Out of 113 retrieved articles, a total of 20 articles were included in our review. These were divided into three categories: seven perception studies, six intervention studies, and seven reliability and validation studies. The first study conducted in Norway indicated a need to improve patient safety culture. Only one intervention study was able to substantially improve patient safety culture. The validity of HSOPSC is supported in most studies. However, one study indicated poor quality in relation to the testing of criteria related to validity. This review is limited to Norwegian healthcare but has several relevant implications across the research field, namely that intervention studies should (1) validate dimensions more carefully, (2) avoid pitfalls related to both factor analysis methods and criteria validity testing, (3) consider integrating structural models into multilevel improvement programs, and (4) benefit from applying different, new versions of HSOPSC developed in Norway.

## 1. Introduction

Patient safety culture consists of the attitudes and routines among healthcare personnel and management that impact patient treatment [1,2]. A positive patient safety culture includes a focus on establishing systems, routines, resources, and infrastructure to reduce risks and errors [3]. Studies indicate an association between a positive patient safety culture and safe patient treatment [3,4,5]. In 2004, the Agency for Healthcare Research and Quality (AHRQ) launched the Hospital Survey on Patient Safety Culture (HSOPSC) version 1.0 to assess patient safety culture in hospitals [1,2]. HSOPSC includes 42 items grouped into 12 composite measures, or composites. Seven dimensions target the unit level, three dimensions target the hospital level, and two composites are outcome measures (overall perception of patient safety and frequency of events reported). HSOPSC also includes two questions that ask respondents to provide an overall grade on patient safety for their work area/unit and to indicate the number of events they reported over the past 12 months. Hospitals have the opportunity to benchmark results against other datasets [6], or potentially against previous baseline measures, to monitor development over time and to evaluate improvement initiatives. All of the measures are illustrated in Figure 1. The survey also includes limited background demographic information (work area/unit, staff position, etc.).

As of September 2020, HSOPSC 1.0 has been administered in 95 countries and translated into 43 languages [7]. In Norway, the first two studies assessing patient safety culture using the HSOPSC were conducted in 2006 and 2008 at Stavanger University Hospital [8,9]. Hence, Norway applied HSOPSC relatively early after the instrument was developed. However, in other sectors and industries, such as the aviation and petroleum sectors and the nuclear industry, assessment of safety culture was already a tradition [10,11,12], so it was certainly not too early to assess safety culture in healthcare settings.

One literature review examined the psychometric properties of several questionnaires designed to measure the safety climate in healthcare [13]. The authors concluded that the HSOPSC covers the most central dimensions of safety culture, and it meets psychometric criteria such as content- and criterion-related validity and internal reliability [13]. Moreover, it was presented as the most comprehensive validated instrument in healthcare, an evaluation which has been supported by several studies [13,14,15,16]. Therefore, HSOPSC is a potentially important tool for improving patient safety [2].

The aims of this study were (1) to review empirical studies using HSOPSC in Norway and (2) to develop recommendations for further research on patient safety culture based on our findings.

## 2. Materials and Methods

Data searches were conducted in Oria, an online catalogue of scientific databases which allows for broad searches across different databases and can be used to find printed and electronic resources at the University Library in Norway. Additional information concerning the sources included from the Oria search is listed in Appendix A. An Oria search includes the Central Discovery Index from ExLibris. This broad search strategy evolved based on discussions with an experienced librarian. Oria searches also include MEDLINE and CHINAL (Appendix A). Searches in Oria were performed using the terms “Hospital Survey on Patient Safety Culture” OR “HSOPSC” AND “Norway”. The searches were conducted between 12 February 2021 and 18 February 2021. In addition, one article was identified by exploring Norwegian researchers (e.g., “Storm”, “Haugen”, “Reierstad”, “Vifladt”) conducting patient safety culture studies. These Norwegian-based authors are listed in Appendix B. This search was conducted using Google Scholar. Moreover, two articles were identified by exploring all papers referring to the first validation study of HSOPSC [16] in Norway using Google Scholar. These were not found in the first search since they were published in books. Hence, several steps were conducted to ensure compliance with the inclusion and exclusion criteria described below.

The study adheres to the PRISMA guidelines for systematic reviews [17]. The inclusion criteria were as follows: (1) the studies were conducted in Norway in a hospital setting, (2) the hospital version of HSOPSC was used (not the nursing home version), and (3) the heading and summary were written in English. The exclusion criterion was nonempirical studies (e.g., study protocols). The study selection PRISMA flowchart is presented in Figure 2.

## 3. Results

A total of 20 articles were included. These were divided into three categories: seven perception studies, six intervention studies, and seven reliability and validation studies.

### 3.1. Perception Studies

Seven studies were categorized as perception studies [8,18,19,20,21,22,23]. Some of these made comparisons with other samples [8,18] as well as repeated measures to monitor change over time [19]. In the first Norwegian study [8], the mean ± standard deviation (M ± SD) was reported. The strongest HSOPSC dimensions were “Teamwork within units” (M ± SD = 3.84 ± 0.60) and “Supervisor/manager expectations and actions promoting safety” (M ± SD = 3.82 ± 0.68). These scores indicate that the mean scores were almost at the level of Agree and were substantially lower than the maximum score of 5. “Organizational management support for safety” had the largest improvement potential, with a mean score (M ± SD = 2.90 ± 0.75) marginally lower than the level of Neither Agree nor Disagree. For the lowest scoring dimension, the standard deviation was also higher, indicating that the perception of this culture dimension is more diverse among staff. Hence, it is enlightening to assess the standard deviation when interpreting the results. The findings indicated that this Norwegian hospital needed to improve patient safety culture and that more or different investments were necessary to achieve this. Moreover, the study also revealed that safety culture dimensions had lower scores compared with those in US hospitals [8], as well as lower scores than in the petroleum industry [18]. Another finding was that safety culture scores are challenging to improve and relatively stable over time [19].

Another study correlated HSOPSC dimensions with burnout and sense of coherence [23]. Findings from this study indicated that a positive safety culture was associated with the absence of burnout and a high ability to cope with stressful situations. As such, the study indicates that safety culture in hospitals is related to employees’ health and stress at work.

### 3.2. Intervention Studies

Six studies involved interventions [24,25,26,27,28,29]. One of these intervention studies reported greater improvement than the others [25]. This study was conducted at Haukeland University Hospital, and HSOPSC measures were collected in 2009, 2010, and 2017. The researchers conducted a stepped wedge cluster randomized controlled trial implementation of the World Health Organization (WHO) Surgical Safety Checklists, combined with the implementation of a broader patient safety program. From 2009 to 2017, significant improvement was found in the following dimensions: “Unit managers’ support to patient safety”, “Continuous improvement”, “Teamwork in unit”, “Error feedback”, “Nonpunitive”, “Hospital managers support to patient safety”, “Teamwork across units”, and “Information handoffs and transitions”. The largest positive changes were related to “Hospital managers’ support to patient safety”, from 2.83 at the baseline in 2009 to a mean score of 3.15 in 2017.

Other intervention studies also reported improvements, but these were generally weaker and reported a shorter intervention period. Aaberg et al. [24,28] found improvement in three HSOPSC dimensions in their two studies: “Teamwork within unit”, “Manager expectations and actions promoting patient safety”, and “Communication openness”. Storm et al. [27] focused their interventions at the interorganizational level. In the hospital part of the study, small improvements were reported for “Overall perceptions of patient safety culture” and “Organizational learning—continuous improvement” [27]. Moreover, one intervention study [29] compared changes in registered nurses’ perception of HSOPSC dimensions in restructured and nonrestructured intensive care units [29] during a four-year period. In this study, restructuring was associated with negative developments in “Manager expectations and actions promoting safety”, “Teamwork within hospital units”, and “Adequate staffing”.

Haugen et al. [26] found significant positive changes in the checklist intervention group for the culture factors “Frequency of events reported” and “Adequate staffing”. Thus, the effects of the intervention were weak since only two dimensions improved.

### 3.3. Reliability and Validation Studies

Seven studies were categorized as reliability or validation studies [16,30,31,32,33,34,35]. Confirmatory factor analysis (CFA) was performed to assess the quality of the measurement model of HSOPSC in the hospital [16,30,31,35] and prehospital settings [31]. The CFA indicated that HSOPSC was a valid and reliable tool for measuring patient safety culture in Norwegian hospitals. Some adjustments were made to the prehospital version, which was labeled PreHSOPSC [31]. Moreover, some items were removed in the development of a short version of HSOPSC, labeled HSOPSC-short [30]. A Short Safety Climate Survey (SSCS) was also developed in Norway, based on HSOPSC, for use in nonhealthcare settings. SSCS is basically similar to the HSOPSC-S, but without the term “patient”. With this adjustment, SSCS can function as a generic instrument to assess safety culture across sectors [30].

One study at Haukeland University Hospital [34] explored the factorial model of HSOPSC dimensions with exploratory factor analysis (EFA) using principal component analysis with Varimax rotation. Since EFA is a dimension reduction method, it was not surprising that the factorial model ended in fewer factors than the original model, namely 10 dimensions instead of 12. However, the study used the original 12-dimensional structure when investigating reliability and conducting benchmarks [34], without confirming the original version of the instrument with CFA. Another study used EFA before using CFA, but this was to develop and validate the abovementioned SSCS and HSOPSC-short [30].

Three of the studies developed and assessed theoretical models with the use of structural equation modeling (SEM), in combination with CFA, or both CFA and EFA [30,32,35]. The first study explored the possibility of a common structural model measuring associations between safety dimensions and safety behavior in the healthcare and petroleum sectors, which was supported [30]. Another SEM study [35] developed and investigated how five selected HSOPSC dimensions influenced safety behavior and overall perceptions of patient safety. Another study [32] investigated a model adapted for the prehospital environment, measuring associations between safety concepts and the outcome dimension “Transitions and handoffs”. These SEM studies are related to, and support, the nomothetical validity of HSOPSC.

One study aimed at testing the criterion-related validity of HSOPSC [33]. Only two medical departments took part in the study, and several HSOPSC dimensions were correlated with adverse events. The Global Trigger Tool (GTT) was used to collect data on adverse events. The study found an inverse association between patient safety culture and adverse events and hence did not support the criterion-related validity of HSOPSC.

## 4. Discussion

Perception studies in Norway have been important for investigating the level of patient safety culture in hospitals, to reveal both strengths and areas for improvement. Studies have revealed that patient safety culture is more positive in US hospitals and the petroleum sector than in Norwegian hospitals [8,18]. This remains a challenge and shows the importance of continuing to focus on improving patient safety culture in Norway.

This review revealed that safety culture dimensions in hospital settings are difficult to improve and can be very stable over time [19]. Moreover, implementing organizational changes, such as restructuring, can even reduce the level of patient safety dimensions [29]. Hence, organizations should never take the challenge of improving and changing patient safety culture lightly. The included intervention studies demonstrated that interventions most often improve very few of the HSOPSC dimensions [24,26,27,28]. Hence, interventions at the team and department levels will normally not improve all of the HSOPSC dimensions. Again, this confirms that realism should be integrated into safety culture improvement efforts. Improving safety culture takes time, is difficult, and can even be hampered by other organizational initiatives [29]. However, one study [25] showed that it is actually possible to change and improve patient safety culture more extensively during an eight-year period. This was achieved through a broad patient safety program, fostering engagement between trust boards, hospital managers, and frontline operating theatre personnel and thus enabling the effective implementation of the Surgical Safety Checklist. This demonstrates the complexity and endurance needed to improve HSOPSC dimensions more thoroughly in hospital settings. Other hospitals should look at this study, as well as the experiences of other industries [9,36], when developing safety improvement programs. Additionally, safety programs should integrate theory and valid measures. Appropriate sampling and data collection methods, units of analysis, levels of data measurement and aggregation, and statistical analyses are also important factors when evaluating such programs and outcomes [37,38].

Validity and reliability are not heavily documented in the included intervention studies. To help to determine the effect of nesting on the results, intraclass correlations (ICCs) can be computed to determine if substantial variation exists between groups compared to variation within groups. ICC describes how strongly units in the same group resemble each other [39], which is relevant to test when conducting interventions. Another challenge concerns aggregation; if the ICC is low, it is a counterargument for aggregating culture scores at the organization level [40]. In turn, this can influence the effects of interventions, making it necessary to add design effects, for instance, when there are many groups with few individuals within each group [39]. These challenges and issues were not integrated nor controlled for in the Norwegian HSOPSC intervention studies.

The first study in Norway [16] showed that the translated Norwegian version of HSOPSC had satisfactory reliability and validity and could be recommended for use in Norwegian hospitals. Further studies should continue to explore the psychometric qualities of HSOPSC in different settings and over time. Since HSOPSC is a standardized instrument, confirmatory factor analysis (CFA) is the appropriate procedure for validation, and not exploratory factor analysis (EFA). If researchers want to test HSOPSC with EFA, then this should be combined with CFA.

However, the most problematic validity concern revealed in this review involves the study aimed at testing the criterion-related validity of HSOPSC [33]. The level of shared variance was not reported in this study, nor was CFA, and only two medical departments took part in the study. One way of handling such data is to aggregate the HSOPSC survey data at the department level before conducting correlations with Global Trigger Tool (GTT) data, which was not done in the study [33]. CFA was not conducted either, nor was, for instance, ICC to test the level of shared variance. Interestingly, Farup [33] also emphasizes other concerns in the study: “Since the GTT never detects all adverse events and the proportion detected is unknown, the results do not indicate the true prevalence of adverse events.” After referring to other studies [41,42,43], Farup also points to the fact that these studies “unveil major problems related to registration of adverse events and demonstrate that the GTT probably is inappropriate for comparisons between units, departments, and hospitals, as an indicator of the true prevalence of adverse events.” Surprisingly, however, Farup did not follow his own recommendations to avoid these pitfalls and Type II error. Hence, the combination of challenges in this study demonstrates the complexity of establishing criterion-related validity for measurement instruments, which is important to focus on when testing criterion validity related to the HSOPSC instrument. Future studies should look carefully at these issues, as well as other recommendations [38], to avoid these pitfalls and to better investigate the criterion-related validity of HSOPSC.

The three studies focused on developing and testing theoretical models with the use of structural equation modeling (SEM) [30,32,35] illustrate the importance of a systems approach to improve safety and specifically patient safety; several factors work in combination and contribute directly and indirectly to the variation in outcome measures, both in the hospital [31] and prehospital settings [32], as well as in a petroleum sector study [30]. Additional research is needed to gain insight into the mechanisms that mediate or moderate improvement efforts for patient safety culture in different settings. We suggest using a multilevel approach emphasizing that all levels in the organization have important safety functions and influence performance at the individual level through behavioral expectancies [9]. Notably, findings from the structural model studies being developed on the basis of theory correspond with the findings from the most successful intervention study in Norway [25]; wider strategic safety initiatives at different levels are needed to improve safety culture more substantially. An interesting future possibility will be to conduct intervention studies building on the structural models being validated and developed in Norway.

## 5. Limitations

This review has some possible limitations. To compensate for the limitations of Oria, three studies were found based on a hand search. With this combined procedure, we assume that all relevant studies related to the inclusion and exclusion criteria were identified. Included studies were limited to Norway. Hence, studies from other countries were not included. Moreover, we did not use any specific method for the synthesis of the results. Due to the studies’ heterogeneity, we did not perform a meta-analysis or a statistical synthesis of findings. Neither did we assess the risk of bias in the included studies. The studies were categorized based on the study approach, which we assumed to be appropriate.

We recommend using the bibliography developed by AHRQ (https://www.ahrq.gov/sops/bibliography/index.html—accessed on 15 March 2021) to learn more about international studies based on HSOPSC. To give an example, 60 studies focusing on improving patient safety culture are listed in this bibliography. Hence, this review does not provide a global assessment of all studies and topics related to HSOPSC. Based on the generic areas discussed in this review, we still believe the results are generalizable beyond Norwegian healthcare settings.

## 6. Conclusions

The aims of this study were to review empirical studies using HSOPSC in Norway and to develop recommendations for further research on patient safety culture based on our findings. Several studies using the HSOPSC have been conducted in Norway, but not at a national level.

Our findings indicate that comprehensive improvement of patient safety culture in hospitals is challenging and may take several years of systematic work. Moreover, experiences from Norway indicate that wider strategic safety initiatives at different levels are needed to improve safety culture more substantially. Research should aim for a more stringent methodological approach. CFA, rather than EFA, should be applied to replicate the dimensional factor structure of HSOPSC. Furthermore, establishing criterion validity is particularly difficult and challenging. We urge future research to avoid possible pitfalls. As a basis for the development of future intervention studies, researchers designing interventions could use the results from the SEM studies to develop more holistic and theoretically sound interventions, including the horizontal and vertical involvement of units and staff. Intervention studies should not take for granted that the reliability and validity of HSOPSC is adequate based on previous studies. It is always a potential pitfall that effective interventions can be evaluated as noneffective if and when psychometric properties of HSOSPC are problematic in certain settings. Researchers can benefit from applying different new versions of HSOPSC that have been developed in Norway: SSCS [30], HOSPSC-short [30], and PreHSOPSC [31]. SSCS has been developed to fit nonhealthcare settings. HSOPSC-short has fewer items and is therefore optimal for combining with other scales, such as work climate dimensions, bullying, job performance, job satisfaction, and work ability [44]. The combination of such scales is highly relevant since safety culture relates to other work factors. PreHSOPSC has been developed to better fit prehospital settings and is probably the best alternative for measuring safety culture in these settings. Hence, this review offers several recommendations for further research that are also relevant for improving safety culture in healthcare.

## Figures and Tables

**Figure 1 ijerph-18-06518-f001:**
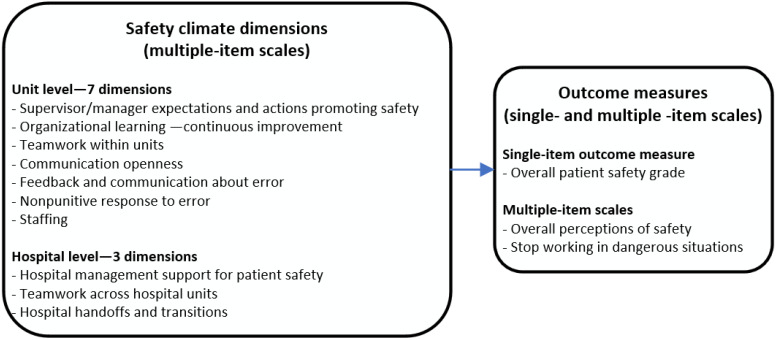
Measurement concepts included in HSOPSC.

**Figure 2 ijerph-18-06518-f002:**
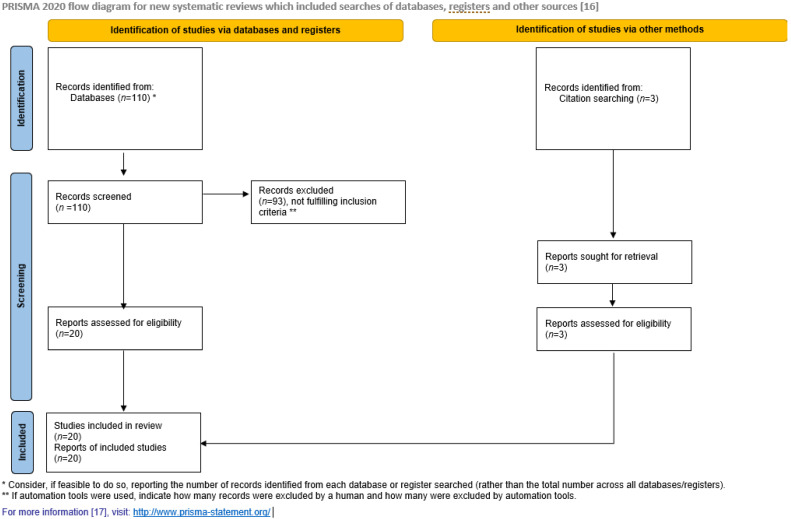
PRISMA 2020 flow diagram for new systematic reviews, which included searches of databases, registers, and other sources.

## Data Availability

Data available on request from the corresponding author.

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
