# Peer review of "Use of the Hospital Survey of Patient Safety Culture in Norwegian Hospitals: A Systematic Review"

_ijerph, 2021, doi:10.3390/ijerph18126518_

Round 1

Reviewer 1 Report

The study is highly interesting to the reader and the community. However, some concerns need to be addressed before allowing publication:

-You selected articles by hand. Why did they not show up via your search? How did you personally identify them? This procedure likely introduced a bias to this study. Please cite as a limitation or better redefine your inclusion criteria in a way that these three (and potentially other) studies are included by your standard search.

-to the unexperienced reader, the ten or twelve dimensions studied should not only be cited but also made available in a figure/table

-Limitations are not at all mentioned in the article. This is the most serious flaw in the manuscript. Please add a section discussing bias (next to the bias mentioned above), generalisability of the findings/the setting (sometimes referred to as external validity, this judgment pertains to the characteristics of the participants in the study, the study setting, the interventions examined, etc.)

Reviewer 2 Report

I think this manuscript is very meaningful.

Summary of this manuscript: In Norway, the Hospital Survey on Patient Safety Culture (HSOPSC) has been widely used since 2006. This review aimed to got an overview of empirical studies using the Hospital Survey on Patient Safety Culture (HSOPSC) in Norway and to develop recommendations for further research on patient safety culture.

Strength: Improving Patient Safety Culture leads to the provision of quality medical care.

Weakness and Improvements are mentioned below:

  1. HSOPSC is the first abbreviation in the introduction. Please correct it to the Hospital Survey on Patient Safety Culture (HSOPSC). [Page 1, Line 40]
  2. Please specify what contents of other sectors and industries are. [Page 1-2, Line 40-41]
  3. I think it would be clearer to change "measure safety climate" to "measure safety climate in healthcare." [Page 2, Line 50]
  4. The aims of this study were 1) to review empirical studies using HSOPSC in Norway, and 2) to develop recommendations for further research on patient safety culture based on our findings. However, the conclusion of this study was “Several studies using the HSOPSC have been conducted in Norway, but not at a national level. Our findings indicate a need for a more stringent methodological approach in several studies. To improve patient safety culture, results indicate that comprehensive improvement will demand several years of systematic work and involve multiple levels in Norwegian hospital settings.” Thus, second aim and conclusion have not matched. [Page 2, Line 59-60, Page 8, 287]
  5. This research should specify for each database its name (such as MEDLINE, CINAHL), the interface or platform through which the database was searched (such as Ovid, EBSCOhost), and the dates of coverage (where this information is provided) in the Preferred Reporting Items for Systematic Reviews and Meta-Analyses (PRISMA) statement [Page 2, Line 63]
  6. Author mentioned that the 20 articles were divided into eight categories based on their 80 approach (see Figure 2). [Page 3, Line 81] However, I think these articles could be divided into three categories. Can one item be expressed as a category? Isn't the third category the reliability and validation of scales (EFA, CFA, and Criterion related validity)? Also, I think "Correlations with Burnout and Sense of Coherence" belong to the Perception study.
  7. The reader cannot understand the meaning of the average score. Please explain what this score means. [Page 4, Line 90-92]
  8. “I think the expression "General," is inappropriate. Please state exactly what was clarified in this study. [Page 6, Line 186; Page 7, Line 221]
  9. This review revealed that safety culture dimensions in the hospital settings are difficult to improve [16]. Organizations should never take easy on the challenge of improving and changing patient safety cultures. Improving safety culture takes time, is difficult, and can even be hampered by other organizational initiatives [25]. One study [21] demonstrates the complexity and endurance needed to improve HSOPSC dimensions more thoroughly in the hospital settings. [Page 6, Line 193-212] These findings are the barrier to improve safety culture dimensions. From these results, in order to achieve "to develop recommendations for further research on patient safety culture," please consider further consideration with reference use research findings in countries other than Norway.

Authors can describe recommendations for further research on patient safety culture by to make strategies to remove these barriers.

  1. If you search PubMed with the following keywords, 96 documents will be hit. “Hospital Survey Patient Safety Culture HSOPSC”

https://pubmed.ncbi.nlm.nih.gov/?term=Hospital+Survey+Patient+Safety+Culture+HSOPSC

65 results by “Hospital Survey Patient Safety Culture HSOPSC improve”

https://pubmed.ncbi.nlm.nih.gov/?term=Hospital+Survey+Patient+Safety+Culture+HSOPSC+improve

I hope you will find it useful.
Thank you very much. 

Round 2

Reviewer 1 Report

The authors are to be applauded for their extensive efforts directed towards the revision of their manuscript - no objections on my end!

Author Response

Reviewer 1

Thank you for the positive constructive feedback during this review process. We are glad you are happy with our responses to your comments.

Comment 1: English language and style are fine/minor spell check required

Response: We have proof-read the final submitted version of our manuscript

Reviewer 2 Report

I think this manuscript has been improved.

Please review and revise based on the reviewer’s comments with coauthors.

Comment 4: Conclusions.

SSC is generic and can be utilized across sectors other than health care.

What is SSC?

This study is a systematic review focused on Patient Safety Culture. It is inconsistent to say "SSC is generic and can be utilized across sectors other than health care" in the conclusion.

I think it would be a clearer conclusion to remove this sentence.

Comment 5:

You can add the following sentence:

Oria is a search engine, which can use to find printed and electronic resources at the University Library in Norway.  

Please be consistent "Oria."

Comment 6:

[Page 4 Line 110]

You can describe below:

The 20 articles were divided into three categories, seven perception studies, six intervention studies, and seven reliability and validation studies.

You do not need Figure 3.

[Page 5 Line 172]

3.3. Reliability and validation studies

You can revise the following sentence. 

Confirmatory Factor Analysis (CFA) was performed to assess the quality of the measurement model of HSOPSC in a hospital [15,29,30,34] and prehospital setting [30]. The CFA indicated HSOPSC was a valid and reliable tool for measuring patient safety culture in Norwegian hospitals.

Comment 7:

[Page 4 Line 118]

Still, the description of the average score is unclear.

You should show the highest score on this dimension (factor) of HSOPSC.

https://www.ahrq.gov/sites/default/files/wysiwyg/professionals/quality-patient-safety/patientsafetyculture/hospital/userguide/hospcult.pdf

https://bmjopen.bmj.com/content/9/5/e028666#T4

Judging from the above literature, the maximum score is 5 points (Strongly agree) on the Likert scale.

Example items are:

We are actively doing things to improve patient safety.

We have patient safety problems in this unit.

Therefore:

  • If respondents answer Always or Strongly agree to a negatively worded item, answers should be recoded from 5 to 1.

Here is an example of computing a percent positive composite score for the composite Overall Perceptions of Safety.

Each factor has a different number of question items. Therefore, we need to calculate the mean factor points (MFP) for the factor.

Teamwork Within Units

A1. People support one another in this unit.

A3. When a lot of work needs to be done quickly, we work together as a team to get the work done.

A4. In this unit, people treat each other with respect.

A11. When one area in this unit gets really busy, others help out.

For example, if you will describe the revised sentences….

The HSOPSC had lower scores than United States (US) hospitals [7]:  US scored a mean ± standard deviation (SD) XX±YY and Norway XX ±YY points. The scores of Norway was lower score in 'Teamwork within units' (mean score on dimension was 3.84 ± standard deviation (SD) NOTE: THIS ALMOST SHOWS level of AGREE), 'Supervisor/manager expectations and actions promoting safety' (mean score on dimension was 3.82±SD NOTE: THIS ALMOST SHOWS level of AGREE).

Especially, 'Organizational management support for safety (mean score on dimension was 2.90±SD NOTE: THIS ALMOST SHOWS level of Neither Agree nor Disagree or near the Disagree).’

This means, Norway hospitals need further continuous improvement.

SD is critically important information to evaluate these data. You have to carefully consider from this information.

Comment 8:

"General,"

This expression still remains on pages 5, 6, 7, and 8. Please search and confirm, and revise.

You can delete this word.

Comment 9:

[Page 8 Line 321]

Please revise the second paragraph of Limitations as follows:

However, this literature review has some possible limitations as systematic reviews.

You don’t need following sentence. I think it would be a clearer review article to remove this sentence.

[Page 8 Line 334]

DELETE: One may argue that reducing our search to studies conducted in a Norwegian hospital setting may limit the generalizability of the findings.

I hope this will be useful to you.

Thank you very much.

Author Response

Thank you for the positive feedback after our improvements. Your additional comments have further improved and supported the development of the paper, and we have done our best to adhere to your advice and suggestions.

Comment 1: Comment 4: Conclusions.

SSC is generic and can be utilized across sectors other than health care.

What is SSC?

This study is a systematic review focused on Patient Safety Culture. It is inconsistent to say "SSC is generic and can be utilized across sectors other than health care" in the conclusion.

I think it would be a clearer conclusion to remove this sentence.

Response: The sentence has been removed

Comment 2: Comment 5:

You can add the following sentence:

Oria is a search engine, which can use to find printed and electronic resources at the University Library in Norway.  Please be consistent "Oria."

Response: We thank the reviewer for this suggestion, and have consequently added the sentence

Comment 3: Comment 6:

[Page 4 Line 110]

You can describe below:

The 20 articles were divided into three categories, seven perception studies, six intervention studies, and seven reliability and validation studies.

Response: We thank the reviewer for this suggestion, and have consequently added the sentence

Comment 4: You do not need Figure 3.

Response: Figure 3 has been removed.

Comment 5: [Page 5 Line 172]

3.3. Reliability and validation studies

You can revise the following sentence. 

Confirmatory Factor Analysis (CFA) was performed to assess the quality of the measurement model of HSOPSC in a hospital [15,29,30,34] and prehospital setting [30]. The CFA indicated HSOPSC was a valid and reliable tool for measuring patient safety culture in Norwegian hospitals.

Response: We thank the reviewer for this input, and have revised the sentence accordingly.

Comment 6: Comment 7:

[Page 4 Line 118]

Still, the description of the average score is unclear.

You should show the highest score on this dimension (factor) of HSOPSC.

https://www.ahrq.gov/sites/default/files/wysiwyg/professionals/quality-patient-safety/patientsafetyculture/hospital/userguide/hospcult.pdf

https://bmjopen.bmj.com/content/9/5/e028666#T4

Judging from the above literature, the maximum score is 5 points (Strongly agree) on the Likert scale.

Example items are:

We are actively doing things to improve patient safety.

We have patient safety problems in this unit.

Therefore:

  • If respondents answer Always or Strongly agree to a negatively worded item, answers should be recoded from 5 to 1.

Here is an example of computing a percent positive composite score for the composite Overall Perceptions of Safety.

Each factor has a different number of question items. Therefore, we need to calculate the mean factor points (MFP) for the factor.

Teamwork Within Units

A1. People support one another in this unit.

A3. When a lot of work needs to be done quickly, we work together as a team to get the work done.

A4. In this unit, people treat each other with respect.

A11. When one area in this unit gets really busy, others help out.

Response: We thank the reviewer for constructive and concrete input. We have revised this explanation accordingly. Please see manuscript changes in chapter 1 and particularly 3.1

3.1. Perception studies

Seven studies were categorized as perception studies [8,18-23]. Some of these did comparisons with other samples [8,18] as well as repeated measures to monitor change over time [19]. In the first Norwegian study [8] mean ± standard deviation (M ± SD) were reported. The strongest HSOPSC dimensions were ‘Teamwork within units’ (M ± SD: 3.84 ± 0.60) and ‘Supervisor/manager expectations and actions promoting safety’ (M ± SD = 3.82 ± 0.68). These scores indicate mean scores were almost at level of Agree, and substantially lower than the maximum score of 5. ‘Organisational management support for safety’ had the largest improvement potential with a mean score (M ± SD = 2.90 ± 0.75) marginally lower than level of Neither Agree nor Disagree. For the lowest scoring dimension, the standard deviation was also higher, indicating that the perception of this culture dimension is more diverse among staff. Hence, it is interesting to also assess the standard deviation when interpreting the results. Findings indicated this Norwegian hospital needed to improve the patient safety culture level, on invest more or different, to be able to achieve this. Moreover, this study also revealed that safety culture dimensions had lower scores compared with US hospitals [8], and also lower than scores in the petroleum industry [18]. Another finding was that safety culture scores are challenging to improve and relatively stable over time [19].
